# LONG-TERM FORECASTING USING TENSOR-TRAIN RNNS

## ABSTRACT

We present Tensor-Train RNN (`TT-RNN`), a novel family of neural sequence architectures for multivariate forecasting in environments with nonlinear dynamics. Long-term forecasting in such systems is highly challenging, since there exist long-term temporal dependencies, higher-order correlations and sensitivity to error propagation. Our proposed tensor recurrent architecture addresses these issues by learning the nonlinear dynamics directly using higher order moments and high-order state transition functions. Furthermore, we decompose the higher-order structure using the tensor-train (TT) decomposition to reduce the number of parameters while preserving the model performance. We theoretically establish the approximation properties of Tensor-Train RNNs for general sequence inputs, and such guarantees are not available for usual RNNs. We also demonstrate significant long-term prediction improvements over general RNN and LSTM architectures on a range of simulated environments with nonlinear dynamics, as well on real-world climate and traffic data.

## 1   INTRODUCTION

One of the central questions in science is forecasting: given the past history, how well can we predict the future? In many domains with complex multivariate correlation structures and nonlinear dynamics, forecasting is highly challenging since the system has long-term temporal dependencies and higher-order dynamics. Examples of such systems abound in science and engineering, from biological neural network activity, fluid turbulence, to climate and traffic systems (see Figure 1). Since current forecasting systems are unable to faithfully represent the higher-order dynamics, they have limited ability for accurate *long-term* forecasting.

Therefore, a key challenge is accurately modeling nonlinear dynamics and obtaining stable long-term predictions, given a dataset of realizations of the dynamics. Here, the forecasting problem can be stated as follows: how can we efficiently learn a model that, given only few initial states, can reliably predict a sequence of future states over a long horizon of $T$ time-steps?

Common approaches to forecasting involve linear time series models such as auto-regressive moving average (ARMA), state space models such as hidden Markov model (HMM), and deep neural networks. We refer readers to a survey on time series forecasting by (Box et al., 2015) and the references therein. A recurrent neural network (RNN), as well as its memory-based extensions such as the LSTM, is a class of models that have achieved good performance on sequence prediction tasks from demand forecasting (Flunkert et al., 2017) to speech recognition (Soltau et al., 2016) and video analysis (LeCun

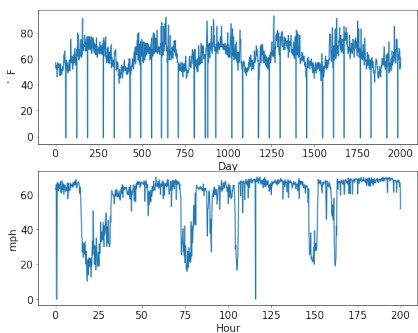

Figure 1: Climate and traffic time series per location. The time-series exhibits long-term temporal correlations, and can be viewed as a realization of highly nonlinear dynamics.

et al., 2015). Although these methods can be effective for short-term, smooth dynamics, neither analytic nor data-driven learning methods tend to generalize well to capturing long-term nonlinear dynamics and predicting them over longer time horizons.

To address this issue, we propose a novel family of tensor-train recurrent neural networks that can learn stable long-term forecasting. These models have two key features: they 1) *explicitly model the higher-order dynamics*, by using a longer history of previous hidden states and high-order state interactions with multiplicative memory units; and 2) they are scalable by using *tensor trains*, a structured low-rank tensor decomposition that greatly reduces the number of model parameters, while mostly preserving the correlation structure of the full-rank model.

In this work, we analyze Tensor-Train RNNs theoretically, and also experimentally validate them over a wide range of forecasting domains. Our contributions can be summarized as follows:

- We describe how `TT-RNNs` encode higher-order non-Markovian dynamics and high-order state interactions. To address the memory issue, we propose a tensor-train (TT) decomposition that makes learning tractable and fast.

- We provide theoretical guarantees for the representation power of `TT-RNNs` for nonlinear dynamics, and obtain the connection between the target dynamics and `TT-RNN` approximation. In contrast, no such theoretical results are known for standard recurrent networks.

- We validate `TT-RNNs` on simulated data and two real-world environments with nonlinear dynamics (climate and traffic). Here, we show that `TT-RNNs` can forecast more accurately for significantly longer time horizons compared to standard RNNs and LSTMs.

## 2 FORECASTING USING TENSOR-TRAIN RNNS

**Forecasting Nonlinear Dynamics**    Our goal is to learn an efficient model $f$ for *sequential multivariate forecasting* in environments with nonlinear dynamics. Such systems are governed by *dynamics* that describe how a system state $\mathbf{x}_t \in \mathbb{R}^d$ evolves using a set of *nonlinear* differential equations:

$$\left\{ \xi^i \left( \mathbf{x}_t, \frac{d\mathbf{x}}{dt}, \frac{d^2\mathbf{x}}{dt^2}, \dots; \phi \right) = 0 \right\}_i, \tag{1}$$

where $\xi^i$ can be an arbitrary (smooth) function of the state $\mathbf{x}_t$ and its derivatives. Continous time dynamics are usually described by differential equations while difference equations are employed for discrete time. In continuous time, a classic example is the first-order Lorenz attractor, whose realizations showcase the "butterfly-effect", a characteristic set of double-spiral orbits. In discrete-time, a non-trivial example is the 1-dimensional Genz dynamics, whose difference equation is:

$$x_{t+1} = \left( c^{-2} + (x_t + w)^2 \right)^{-1}, \quad c, w \in [0, 1], \tag{2}$$

where $x_t$ denotes the system state at time $t$ and $c, w$ are the parameters. Due to the nonlinear nature of the dynamics, such systems exhibit higher-order correlations, long-term dependencies and sensitivity to error propagation, and thus form a challenging setting for learning. Given a sequence of initial states $\mathbf{x}_0 \dots \mathbf{x}_t$, the forecasting problem aims to learn a model $f$

$$f : (\mathbf{x}_0 \dots \mathbf{x}_t) \mapsto (\mathbf{y}_t \dots \mathbf{y}_T), \quad \mathbf{y}_t = \mathbf{x}_{t+1}, \tag{3}$$

that outputs a sequence of future states $\mathbf{x}_{t+1} \dots \mathbf{x}_T$. Hence, accurately approximating the dynamics $\xi$ is critical to learning a good forecasting model $f$ and accurately predicting for long time horizons.

**First-order Markov Models**    In deep learning, common approaches for modeling dynamics usually employ first-order hidden-state models, such as recurrent neural networks (RNNs). An RNN with a single RNN cell recursively computes the output $\mathbf{y}_t$ from a hidden state $\mathbf{h}_t$ using:

$$\mathbf{h}_t = f(\mathbf{x}_t, \mathbf{h}_{t-1}; \theta), \quad \mathbf{y}_t = g(\mathbf{h}_t; \theta), \tag{4}$$

where $f$ is the state transition function, $g$ is the output function and $\theta$ are model parameters. An RNN only considers the most recent hidden state in its state transition function. A common parametrization scheme for (4) is a nonlinear activation function applied to a linear map of $\mathbf{x}_t$ and $\mathbf{h}_{t-1}$ as:

$$\mathbf{h}_t = f(W^{hx}\mathbf{x}_t + W^{hh}\mathbf{h}_{t-1} + \mathbf{b}^h), \quad \mathbf{x}_{t+1} = W^{xh}\mathbf{h}_t + \mathbf{b}^x, \tag{5}$$

where $f$ is the activation function (e.g. sigmoid, $\tanh$) for the state transition, $W^{hx}, W^{xh}$ and $W^{hh}$ are transition weight matrices and $\mathbf{b}^h, \mathbf{b}^x$ are biases. RNNs have many different variations, including

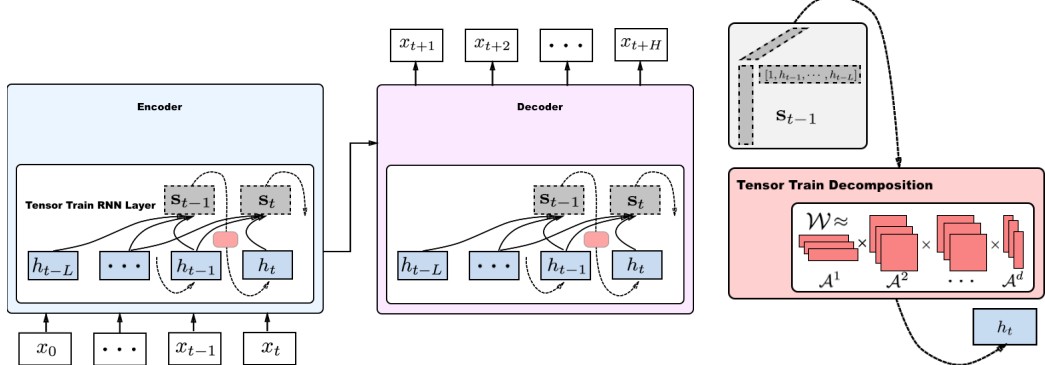

Figure 2: Tensor-train recurrent cells within a seq2seq model. Both encoder (blue) and decoder (pink) contain tensor-train recurrent cells (red) with high-order hidden states.

Figure 3: Tensor-train cell with factorized hidden states using tensor-train model.

LSTMs (Hochreiter & Schmidhuber, 1997) and GRUs (Chung et al., 2014). For instance, LSTM cells use a memory-state, which mitigate the "exploding gradient" problem and allow RNNs to propagate information over longer time horizons. Although RNNs are very expressive, they compute $\mathbf{h}_t$ only using the previous state $\mathbf{h}_{t-1}$ and input $\mathbf{x}_t$. Such models do not explicitly model higher-order dynamics and only implicitly model long-term dependencies between all historical states $\mathbf{h}_0 \ldots \mathbf{h}_t$, which limits their forecasting effectiveness in environments with nonlinear dynamics.

## 2.1 TENSORIZED RECURRENT NEURAL NETWORKS

To effectively learn nonlinear dynamics, we propose Tensor-Train RNNs, or `TT-RNNs`, a class of higher-order models that can be viewed as a higher-order generalization of RNNs. We developed `TT-RNNs` with two goals in mind: explicitly modeling 1) $L$-order Markov processes with $L$ steps of temporal memory and 2) polynomial interactions between the hidden states $\mathbf{h}.$ and $\mathbf{x}_t$.

First, we consider longer "history": we keep length $L$ historic states: $\mathbf{h}_t, \cdots, \mathbf{h}_{t-L}$:
$$\mathbf{h}_t = f(\mathbf{x}_t, \mathbf{h}_{t-1}, \cdots, \mathbf{h}_{t-L}; \theta)$$
where $f$ is an activation function. In principle, early work (Giles et al., 1989) has shown that with a large enough hidden state size, such recurrent structures are capable of approximating any dynamics.

Second, to learn the nonlinear dynamics $\xi$ efficiently, we also use higher-order moments to approximate the state transition function. We construct a higher-order transition *tensor* by modeling a degree $P$ polynomial interaction between hidden states. Hence, the `TT-RNN` with standard RNN cell is defined by:
$$[\mathbf{h}_t]_\alpha = f(W_\alpha^{hx}\mathbf{x}_t + \sum_{i_1,\cdots,i_p} \mathcal{W}_{\alpha i_1 \cdots i_P} \underbrace{\mathbf{s}_{t-1;i_1} \otimes \cdots \otimes \mathbf{s}_{t-1;i_p}}_{P}) \tag{6}$$
where $\alpha$ index the hidden dimension, $i.$ index historic hidden states and $P$ is the polynomial degree. Here, we defined the $L$-lag hidden state as:
$$\mathbf{s}_{t-1}^T = [1 \quad \mathbf{h}_{t-1}^\top \quad \ldots \quad \mathbf{h}_{t-L}^\top]$$
We included the bias unit 1 to model all possible polynomial expansions up to order $P$ in a compact form. The `TT-RNN` with LSTM cell, or "TLSTM", is defined analogously as:
$$\begin{bmatrix} \mathbf{i}_t \\ \mathbf{g}_t \\ \mathbf{f}_t \\ \mathbf{o}_t \end{bmatrix}_\alpha = \sigma\left(W_\alpha^{hx}\mathbf{x}_t + \sum_{i_1,\cdots,i_p} \mathcal{W}_{\alpha i_1 \cdots i_P} \underbrace{\mathbf{s}_{t-1;i_1} \otimes \cdots \otimes \mathbf{s}_{t-1;i_p}}_{P}\right),$$
$$\mathbf{c}_t = \mathbf{c}_{t-1} \circ \mathbf{f}_t + \mathbf{i}_t \circ \mathbf{g}_t, \quad \mathbf{h}_t = \mathbf{c}_t \circ \mathbf{o}_t \tag{7}$$
where $\circ$ denotes the Hadamard product. Note that the bias units are again included. `TT-RNN` serves as a module for sequence-to-sequence (Seq2Seq) framework (Sutskever et al., 2014), which consists of an encoder-decoder pair (see Figure 2). We use tensor-train recurrent cells both the encoder and decoder. The encoder receives the initial states and the decoder predicts $x_{t+1}, \ldots, x_T$. For each timestep $t$, the decoder uses its previous prediction $\mathbf{y}_t$ as an input.

## 2.2 Tensor-train Networks

Unfortunately, due to the "curse of dimensionality", the number of parameters in $\mathcal{W}_\alpha$ with hidden size $H$ grows exponentially as $O(HL^P)$, which makes the high-order model prohibitively large to train. To overcome this difficulty, we utilize *tensor networks* to approximate the weight tensor. Such networks encode a structural decomposition of tensors into low-dimensional components and have been shown to provide the most general approximation to smooth tensors (Orús, 2014). The most commonly used tensor networks are *linear tensor networks* (LTN), also known as *tensor-trains* in numerical analysis or *matrix-product states* in quantum physics (Oseledets, 2011).

A tensor train model decomposes a $P$-dimensional tensor $\mathcal{W}$ into a network of sparsely connected low-dimensional tensors $\{\mathcal{A}^d \in \mathbb{R}^{r_{d-1} \times n_d \times r_d}\}$ as:

$$\mathcal{W}_{i_1 \cdots i_P} = \sum_{\alpha_1 \cdots \alpha_{P-1}} \mathcal{A}^1_{\alpha_0 i_1 \alpha_1} \mathcal{A}^2_{\alpha_1 i_2 \alpha_2} \cdots \mathcal{A}^P_{\alpha_{P-1} i_P \alpha_P}, \quad \alpha_0 = \alpha_P = 1$$

as depicted in Figure (3). When $r_0 = r_P = 1$ the $\{r_d\}$ are called the tensor-train rank. With tensor-train, we can reduce the number of parameters of TT-RNN from $(HL+1)^P$ to $(HL+1)R^2 P$, with $R = \max_d r_d$ as the upper bound on the tensor-train rank. Thus, a major benefit of tensor-train is that they *do not* suffer from the curse of dimensionality, which is in sharp contrast to many classical tensor decompositions, such as the Tucker decomposition.

# 3 Approximation results for TT-RNN

A significant benefit of using tensor-trains is that we can theoretically characterize the representation power of tensor-train neural networks for approximating high-dimensional functions. We do so by analyzing a class of functions that satisfies some regularity condition. For such functions, tensor-train decompositions preserve weak differentiability and yield a compact representation. We combine this property with neural network estimation theory to bound the approximation error for TT-RNN with one hidden layer in terms of: 1) the regularity of the target function $f$, 2) the dimension of the input space, 3) the tensor train rank and 4) the order of the tensor.

In the context of TT-RNN, the target function $f(\mathbf{x})$, with $\mathbf{x} = \mathbf{s} \otimes \ldots \otimes \mathbf{s}$, describes the state transitions of the system dynamics, as in (6). Let us assume that $f(\mathbf{x})$ is a Sobolev function: $f \in \mathcal{H}^k_\mu$, defined on the input space $\mathcal{I} = I_1 \times I_2 \times \cdots I_d$, where each $I_i$ is a set of vectors. The space $\mathcal{H}^k_\mu$ is defined as the functions that have bounded derivatives up to some order $k$ and are $L_\mu$-integrable:

$$\mathcal{H}^k_\mu = \left\{ f \in L_\mu(I) : \sum_{i \leq k} \|D^{(i)} f\|^2 < +\infty \right\}, \tag{8}$$

where $D^{(i)} f$ is the $i$-th weak derivative of $f$ and $\mu \geq 0$.[1] Any Sobolev function admits a Schmidt decomposition: $f(\cdot) = \sum_{i=0}^{\infty} \sqrt{\lambda(i)} \gamma(\cdot; i) \otimes \phi(i; \cdot)$, where $\{\lambda\}$ are the eigenvalues and $\{\gamma\}, \{\phi\}$ are the associated eigenfunctions. Hence, we can decompose the target function $f \in \mathcal{H}^k_\mu$ as:

$$f(\mathbf{x}) = \sum_{\alpha_0, \cdots, \alpha_d = 1}^{\infty} \mathcal{A}^1(\alpha_0, x_1, \alpha_1) \cdots \mathcal{A}^d(\alpha_{d-1}, x_d, \alpha_d), \tag{9}$$

where $\{\mathcal{A}^d(\alpha_{d-1}, \cdot, \alpha_d)\}$ are basis functions $\{\mathcal{A}^d(\alpha_{d-1}, x_d, \alpha_d) = \sqrt{\lambda_{d-1}(\alpha_{d-1})} \phi(\alpha_{d-1}; x_d)\}$, satisfying $\langle \mathcal{A}^d(i, \cdot, m), \mathcal{A}^d(i, \cdot, m) \rangle = \delta_{mn}$. We can truncate (13) to a low dimensional subspace ($\mathbf{r} < \infty$), and obtain the *functional tensor-train (FTT)* approximation of the target function $f$:

$$f_{TT}(\mathbf{x}) = \sum_{\alpha_0, \cdots, \alpha_d = 1}^{\mathbf{r}} \mathcal{A}^1(\alpha_0, x_1, \alpha_1) \cdots \mathcal{A}^d(\alpha_{d-1}, x_d, \alpha_d), \tag{10}$$

In practice, TT-RNN implements a polynomial expansion of the state $\mathbf{s}$ as in (6), using powers $[\mathbf{s}, \mathbf{s}^{\otimes 2}, \cdots, \mathbf{s}^{\otimes p}]$ to approximate $f_{TT}$, where $p$ is the degree of the polynomial. We can then bound the approximation error using TT-RNN, viewed as a one-layer hidden neural network:

---

[1]A weak derivative generalizes the derivative concept for (non)-differentiable functions and is implicitly defined as: e.g. $v \in L^1([a, b])$ is a weak derivative of $u \in L^1([a, b])$ if for all smooth $\varphi$ with $\varphi(a) = \varphi(b) = 0$: $\int_a^b u(t) \varphi'(t) = -\int_a^b v(t) \varphi(t)$.

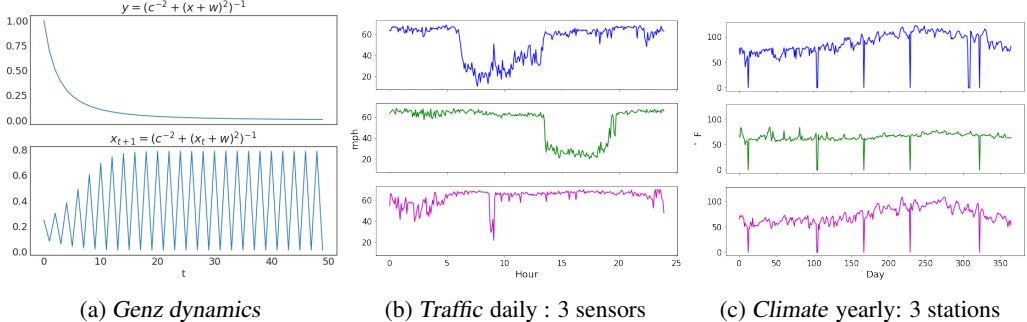

|(a) *Genz dynamics* | (b) *Traffic* daily : 3 sensors | (c) *Climate* yearly: 3 stations |

Figure 4: Data visualizations: (a) Genz dynamics, (b) traffic data, (c) climate data.

**Theorem 3.1.** *Let the state transition function $f \in \mathcal{H}_\mu^k$ be a Hölder continuous function defined on a input domain $\mathbf{I} = I_1 \times \cdots \times I_d$, with bounded derivatives up to order $k$ and finite Fourier magnitude distribution $C_f$. Then a single layer Tensor Train RNN can approximate $f$ with an estimation error of $\epsilon$ using with $h$ hidden units:*

$$h \le \frac{C_f^2}{\epsilon}(d-1)\frac{(r+1)^{-(k-1)}}{(k-1)} + \frac{C_f^2}{\epsilon}C(k)p^{-k}$$

*where $C_f = \int |\omega|_1 |\hat{f}(\omega)d\omega|$, $d$ is the size of the state space, $r$ is the tensor-train rank and $p$ is the degree of high-order polynomials i.e., the order of tensor.*

For the full proof, see the Appendix. From this theorem we see: 1) if the target $f$ becomes smoother, it is easier to approximate and 2) polynomial interactions are more efficient than linear ones in the large rank region: if the polynomial order increases, we require fewer hidden units $n$. This result applies to the full family of `TT-RNN`s, including those using vanilla RNN or LSTM as the recurrent cell, as long as we are given a state transitions $(\mathbf{x}_t, \mathbf{s}_t) \mapsto \mathbf{s}_{t+1}$ (e.g. the state transition function learned by the encoder).

## 4 EXPERIMENTS

### 4.1 DATASETS

We validated the accuracy and efficiency of `TT-RNN` on one synthetic and two real-world datasets, as described below; Detailed preprocessing and data statistics are deferred to the Appendix.

**Genz dynamics** The Genz "product peak" (see Figure 4 a) is one of the Genz functions (Genz, 1984), which are often used as a basis for high-dimensional function approximation. In particular, (Bigoni et al., 2016) used them to analyze tensor-train decompositions. We generated $10,000$ samples of length $100$ using (2) with $w = 0.5, c = 1.0$ and random initial points.

**Traffic** The traffic data (see Figure 4 b) of Los Angeles County highway network is collected from California department of transportation `http://pems.dot.ca.gov/`. The prediction task is to predict the speed readings for $15$ locations across LA, aggregated every $5$ minutes. After upsampling and processing the data for missing values, we obtained $8,784$ sequences of length $288$.

**Climate** The climate data (see Figure 4 c) is collected from the U.S. Historical Climatology Network (USHCN) (`http://cdiac.ornl.gov/ftp/ushcn_daily/`). The prediction task is to predict the daily maximum temperature for $15$ stations. The data spans approximately $124$ years. After preprocessing, we obtained $6,954$ sequences of length $366$.

### 4.2 LONG-TERM FORECASTING EVALUATION

**Experimental Setup** To validate that `TT-RNN`s effectively perform long-term forecasting task in (3), we experiment with a seq2seq architecture with `TT-RNN` using LSTM as recurrent cells

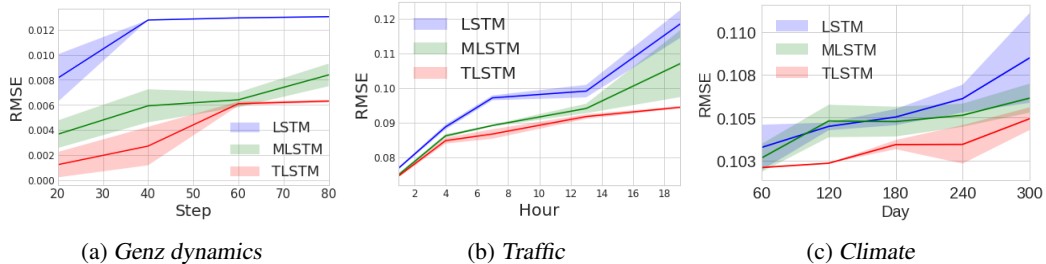

(a) *Genz dynamics*  (b) *Traffic*  (c) *Climate*

Figure 5: Forecasting RMSE for *Genz dynamics* and real world *traffic*, *climate* time series for varying forecasting horizon for LSTM, MLSTM, and TLSTM.

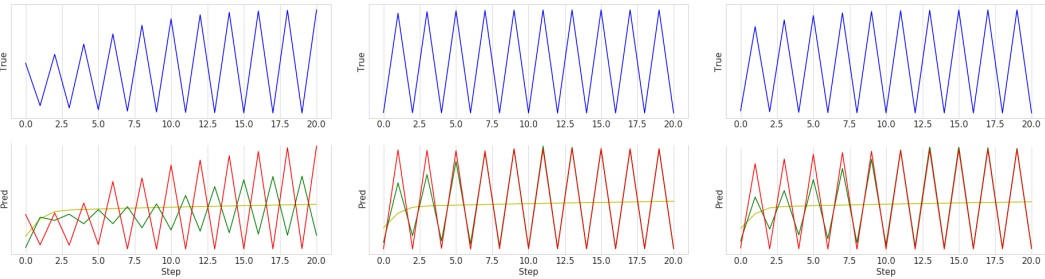

Figure 6: Model prediction for three realizations with different intiial conditions for Genz dynamics "product peak". Top (blue): ground truth. Bottom: model predictions for LSTM (green) and TLSTM (red). TLSTM perfectly captures the Genz oscillations, whereas the LSTM fails to do so (left) or only approaches the ground truth towards the end (middle and right).

(TLSTM). For all experiments, we used an initial sequence of length $t_0$ as input and varied the forecasting horizon $T$. We trained all models using stochastic gradient descent on the length-$T$ sequence regression loss $L(y, \hat{y}) = \sum_{t=1}^{T} ||\hat{y}_t - y_t||_2^2$, where $y_t = x_{t+1}, \hat{y}_t$ are the ground truth and model prediction respectively. For more details on training and hyperparameters, see the Appendix.

We compared `TT-RNN` against 2 set of natural baselines: 1st-order RNN (vanilla RNN, LSTM), and matrix RNNs (vanilla MRNN, MLSTM), which use matrix products of multiple hidden states without factorization (Soltani & Jiang, 2016)). We observed that `TT-RNN` with RNN cells outperforms vanilla RNN and MRNN, but using LSTM cells performs best in all experiments. We also evaluated the classic ARIMA time series model and observed that it performs $\sim 5\%$ worse than LSTM.

**Long-term Accuracy**  For *traffic*, we forecast up to 18 hours ahead with 5 hours as initial inputs. For *climate*, we forecast up to 300 days ahead given 60 days of initial observations. For *Genz dynamics*, we forecast for 80 steps given 5 initial steps. All results are averages over 3 runs.

We now present the long-term forecasting accuracy of TLSTM in nonlinear systems. Figure 5 shows the test prediction error (in RMSE) for varying forecasting horizons for different datasets. We can see that TLSTM notably outperforms all baselines on all datasets in this setting. In particular, TLSTM is more robust to long-term error propagation. We observe two salient benefits of using `TT-RNN`s over the unfactorized models. First, MRNN and MLSTM can suffer from overfitting as the number of weights increases. Second, on *traffic*, unfactorized models also show considerable instability in their long-term predictions. These results suggest that tensor-train neural networks learn more stable representations that generalize better for long-term horizons.

**Visualization of Predictions**  To get intuition for the learned models, we visualize the best performing TLSTM and baselines in Figure 6 for the Genz function "corner-peak" and the state-transition function. We can see that TLSTM can almost perfectly recover the original function, while LSTM and MLSTM only correctly predict the mean. These baselines cannot capture the dynamics fully, often predicting an incorrect range and phase for the dynamics.

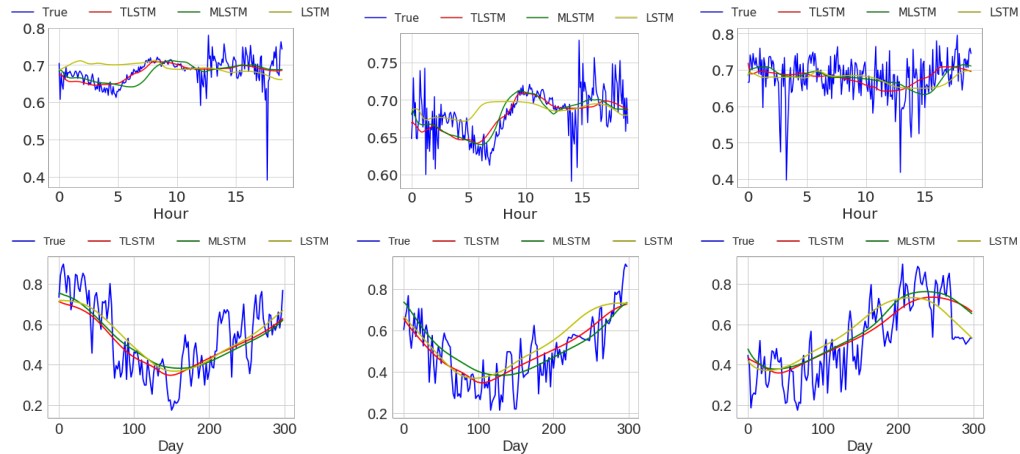

Figure 7: Top: 18 hour ahead predictions for hourly *traffic* time series given 5 hour as input for LSTM, MLSTM and TLSTM. Bottom: 300 days ahead predictions for daily *climate* time series given 2 month observations as input for LSTM, MLSTM and TLSTM.

**TLSTM Prediction Error (RMSE $\times 10^{-2}$)**

| Tensor rank $r$ | 2 | 4 | 8 | 16 |
|---|---|---|---|---|
| Genz ($T = 95$) | **0.82** | 0.93 | 1.01 | 1.01 |
| Traffic ($T = 67$) | 9.17 | **9.11** | 9.32 | 9.31 |
| Climate ($T = 360$) | 10.55 | **10.25** | 10.51 | 10.63 |

**TLSTM Traffic Prediction Error (RMSE $\times 10^{-2}$)**

| Number of lags $L$ | 2 | 4 | 5 | 6 |
|---|---|---|---|---|
| $T = 12$ | **7.38** | 7.41 | 7.43 | 7.41 |
| $T = 84$ | **8.97** | 9.31 | 9.38 | 9.01 |
| $T = 156$ | 9.49 | 9.32 | 9.48 | **9.31** |
| $T = 228$ | 10.19 | 9.63 | **9.58** | 9.94 |

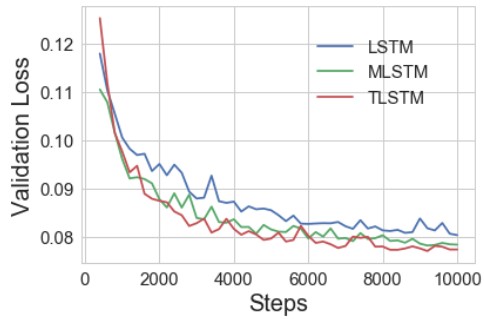

Table 1: TLSTM performance for various tensor-train hyperparameters. Top: varying tensor rank $r$ with $L = 3$. Bottom: varying number of lags $L$ and prediction horizon $T$.

Figure 8: Training speed evaluation: validation loss versus steps for the models with the best long-term forecasting accuracy.

In Figure 7 we show predictions for the real world traffic and climate dataset. We can see that the TLSTM corresponds significantly better with ground truth in long-term forecasting. As the ground truth time series is highly chaotic and noisy, LSTM often deviates from the general trend. While both MLSTM and TLSTM can correctly learn the trend, TLSTM captures more detailed curvatures due to the inherent high-order structure.

**Speed Performance Trade-off**   We now investigate potential trade-offs between accuracy and computation. Figure 8 displays the validation loss with respect to the number of steps, for the best performing models on long-term forecasting. We see that `TT-RNNs` converge significantly faster than other models, and achieve lower validation-loss. This suggests that `TT-RNN` has a more efficient representation of the nonlinear dynamics, and can learn much faster as a result.

**Hyper-parameter Analysis**   The TLSTM model is equipped with a set of hyper-parameters, such as tensor-train rank and the number of lags. We perform a random grid search over these hyper-parameters and showcase the results in Table 1. In the top row, we report the prediction RMSE for the largest forecasting horizon w.r.t tensor ranks for all the datasets with lag 3. When the rank is too low, the model does not have enough capacity to capture non-linear dynamics. when the rank is too high, the model starts to overfit. In the bottom row, we report the effect of changing lags (degree of orders in Markovian dynamics). For each setting, the best $r$ is determined by cross-validation. For different forecasting horizon, the best lag value also varies.

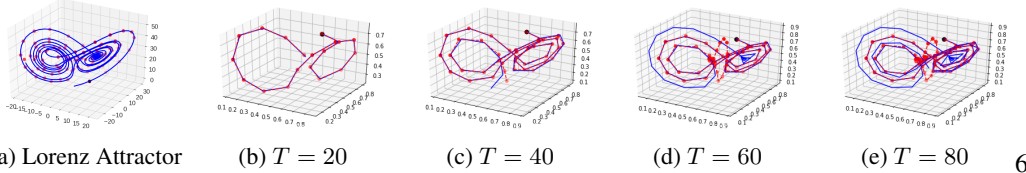

|            |            |            |            |            |
| :--------: | :--------: | :--------: | :--------: | :--------: |
| (a) Lorenz Attractor | (b) $T = 20$ | (c) $T = 40$ | (d) $T = 60$ | (e) $T = 80$ |

Figure 9: a Lorenz attraction with dynamics (blue) and sampled data (red). b, c, d ,e TLSTM long-term predictions for different forecasting horizons $T$ versus the ground truth (blue). TLSTM shows consistent predictions over increasing horizons $T$.

**Chaotic Nonlinear Dynamics** We have also evaluated `TT-RNN` on long-term forecasting for *chaotic* dynamics, such as the Lorenz dynamics (see Figure 9a). Such dynamics are highly sensitive to input perturbations: two close points can move exponentially far apart under the dynamics. This makes long-term forecasting highly challenging, as small errors can lead to catastrophic long-term errors. Figure 9 shows that `TT-RNN` can predict up to $T = 40$ steps into the future, but diverges quickly beyond that. We have found no state-of-the-art prediction model is stable in this setting.

## 5    RELATED WORK

Classic work in time series forecasting has studied auto-regressive models, such as the ARMA or ARIMA model (Box et al., 2015), which model a process $x(t)$ linearly, and so do not capture non-linear dynamics. Our method contrasts with this by explicitly modeling higher-order dependencies. Using neural networks to model time series has a long history. More recently, they have been applied to room temperature prediction, weather forecasting, traffic prediction and other domains. We refer to (Schmidhuber, 2015) for a detailed overview of the relevant literature.

From a modeling perspective, (Giles et al., 1989) considers a *high-order RNN* to simulate a deterministic finite state machine and recognize regular grammars. This work considers a second order mapping from inputs $x(t)$ and hidden states $h(t)$ to the next state. However, this model only considers the most recent state and is limited to two-way interactions. (Sutskever et al., 2011) proposes *multiplicative RNN* that allow each hidden state to specify a different factorized hidden-to-hidden weight matrix. A similar approach also appears in (Soltani & Jiang, 2016), but without the factorization. Our method can be seen as an efficient generalization of these works. Moreover, hierarchical RNNs have been used to model sequential data at multiple resolutions, e.g. to learn both short-term and long-term human behavior (Zheng et al., 2016).

Tensor methods have tight connections with neural networks. For example, (Cohen et al., 2016) shows convolutional neural networks have equivalence to hierarchical tensor factorizations. (Novikov et al., 2015; Yang et al., 2017) employs tensor-train to compress large neural networks and reduce the number of weights. (Yang et al., 2017) forms tensors from reshaping inputs and decomposes the input-output weights. Our model forms tensors from high-order hidden states and decomposes the hidden-output weights. (Stoudenmire & Schwab, 2016) propose to parameterizes the supervised learning models with matrix-product states for image classification. This work however, to the best of our knowledge, is the first work to consider tensor networks in RNNS for sequential prediction tasks for learning in environments with nonlinear dynamics.

## 6    CONCLUSION AND DISCUSSION

In this work, we considered forecasting under nonlinear dynamics.We propose a novel class of RNNs – `TT-RNN`. We provide approximation guarantees for `TT-RNN` and characterize its representation power. We demonstrate the benefits of `TT-RNN` to forecast accurately for significantly longer time horizon in both synthetic and real-world multivariate time series data.

As we observed, chaotic dynamics still present a significant challenge to any sequential prediction model. Hence, it would be interesting to study how to learn robust models for chaotic dynamics. In other sequential prediction settings, such as natural language processing, there does not (or is not known to) exist a succinct analytical description of the data-generating process. It would be interesting to further investigate the effectiveness of `TT-RNN`s in such domains as well.

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

## 7    APPENDIX

### 7.1    THEORETICAL ANALYSIS

We provide theoretical guarantees for the proposed `TT-RNN` model by analyzing a class of functions that satisfy some regularity condition. For such functions, tensor-train decompositions preserve weak differentiability and yield a compact representation. We combine this property with neural network estimation theory to bound the approximation error for `TT-RNN` with one hidden layer, in terms of: 1) the regularity of the target function $f$, 2) the dimension of the input space, and 3) the tensor train rank.

In the context of `TT-RNN`, the target function $f(\mathbf{x})$ with $\mathbf{x} = \mathbf{s} \otimes \ldots \otimes \mathbf{s}$, is the system dynamics that describes state transitions, as in (6). Let us assume that $f(\mathbf{x})$ is a Sobolev function: $f \in \mathcal{H}_\mu^k$, defined on the input space $\mathcal{I} = I_1 \times I_2 \times \cdots I_d$, where each $I_i$ is a set of vectors. The space $\mathcal{H}_\mu^k$ is defined as the set of functions that have bounded derivatives up to some order $k$ and are $L_\mu$-integrable:

$$\mathcal{H}_\mu^k = \left\{ f \in L_\mu^2(I) : \sum_{i \leq k} \|D^{(i)} f\|^2 < +\infty \right\}, \tag{11}$$

where $D^{(i)} f$ is the $i$-th weak derivative of $f$ and $\mu \geq 0$.[2]

Any Sobolev function admits a Schmidt decomposition: $f(\cdot) = \sum_{i=0}^{\infty} \sqrt{\lambda(i)} \gamma(\cdot; i) \otimes \phi(i; \cdot)$, where $\{\lambda\}$ are the eigenvalues and $\{\gamma\}, \{\phi\}$ are the associated eigenfunctions. Hence, we can decompose the target function $f \in \mathcal{H}_\mu^k$ as:

$$f(\mathbf{x}) = \sum_{\alpha_0, \cdots, \alpha_d = 1}^{\infty} \mathcal{A}^1(\alpha_0, x_1, \alpha_1) \cdots \mathcal{A}^d(\alpha_{d-1}, x_d, \alpha_d), \tag{12}$$

where $\{\mathcal{A}^d(\alpha_{d-1}, \cdot, \alpha_d)\}$ are basis functions $\{\mathcal{A}^d(\alpha_{d-1}, x_d, \alpha_d)\} = \sqrt{\lambda_{d-1}(\alpha_{d-1})} \phi(\alpha_{d-1}; x_d)\}$, satisfying $\langle \mathcal{A}^d(i, \cdot, m), \mathcal{A}^d(i, \cdot, m) \rangle = \delta_{mn}$. We can truncate Eqn 13 to a low dimensional subspace ($\mathbf{r} < \infty$), and obtain the *functional tensor-train (FTT)* approximation of the target function $f$:

$$f_{TT}(\mathbf{x}) = \sum_{\alpha_0, \cdots, \alpha_d = 1}^{\mathbf{r}} \mathcal{A}^1(\alpha_0, x_1, \alpha_1) \cdots \mathcal{A}^d(\alpha_{d-1}, x_d, \alpha_d) \tag{13}$$

.

FTT approximation in Eqn 13 projects the target function to a subspace with finite basis. And the approximation error can be bounded using the following Lemma:

**Lemma 7.1** (FTT Approximation Bigoni et al. (2016)). *Let $f \in \mathcal{H}_\mu^k$ be a Hölder continuous function, defined on a bounded domain $\mathbf{I} = I_1 \times \cdots \times I_d \subset \mathbb{R}^d$ with exponent $\alpha > 1/2$, the FTT approximation error can be upper bounded as*

$$\|f - f_{TT}\|^2 \leq \|f\|^2 (d-1) \frac{(r+1)^{-(k-1)}}{(k-1)} \tag{14}$$

*for $r \geq 1$ and*

$$\lim_{r \to \infty} \|f_{TT} - f\|^2 = 0 \tag{15}$$

*for $k > 1$*

Lemma 7.1 relates the approximation error to the dimension $d$, tensor-train rank $r$, and the regularity of the target function $k$. In practice, `TT-RNN` implements a polynomial expansion of the input states $\mathbf{s}$, using powers $[\mathbf{s}, \mathbf{s}^{\otimes 2}, \cdots, \mathbf{s}^{\otimes p}]$ to approximate $f_{TT}$, where $p$ is the degree of the polynomial. We can further use the classic spectral approximation theory to connect the `TT-RNN` structure with the degree of the polynomial, i.e., the order of the tensor. Let $I_1 \times \cdots \times I_d = \mathbf{I} \subset \mathbb{R}^d$. Given a function $f$ and its polynomial expansion $P_{TT}$, the approximation error is therefore bounded by:

---

[2]A weak derivative generalizes the derivative concept for (non)-differentiable functions and is implicitly defined as: e.g. $v \in L^1([a, b])$ is a weak derivative of $u \in L^1([a, b])$ if for all smooth $\varphi$ with $\varphi(a) = \varphi(b) = 0$: $\int_a^b u(t) \varphi'(t) = -\int_a^b v(t) \varphi(t)$.

**Lemma 7.2** (Polynomial Approximation). *Let $f \in \mathcal{H}_\mu^k$ for $k > 0$. Let $P$ be the approximating polynomial with degree $p$, Then*

$$\|f - P_N f\| \leq C(k) p^{-k} |f|_{k,\mu}$$

Here $|f|_{k,\mu}^2 = \sum_{|i|=k} \|D^{(i)} f\|^2$ is the semi-norm of the space $\mathcal{H}_\mu^k$. $C(k)$ is the coefficient of the spectral expansion. By definition, $\mathcal{H}_\mu^k$ is equipped with a norm $\|f\|_{k,\mu}^2 = \sum_{|i| \leq k} \|D^{(i)} f\|^2$ and a semi-norm $|f|_{k,\mu}^2 = \sum_{|i|=k} \|D^{(i)} f\|^2$. For notation simplicity, we muted the subscript $\mu$ and used $\|\cdot\|$ for $\|\cdot\|_{L_\mu}$.

So far, we have obtained the tensor-train approximation error with the regularity of the target function $f$. Next we will connect the tensor-train approximation and the estimation error of neural networks with one layer hidden units. Given a neural network with one hidden layer and sigmoid activation function, following Lemma describes the classic result of describes the error between a target function $f$ and the single hidden-layer neural network that approximates it best:

**Lemma 7.3** (NN Approximation Barron (1993)). *Given a function $f$ with finite Fourier magnitude distribution $C_f$, there exists a neural network of $n$ hidden units $f_n$, such that*

$$\|f - f_n\| \leq \frac{C_f}{\sqrt{n}} \tag{16}$$

*where $C_f = \int |\omega|_1 |\hat{f}(\omega)| d\omega$ with Fourier representation $f(x) = \int e^{i\omega x} \hat{f}(\omega) d\omega$.*

We can now generalize Barron's approximation lemma 7.3 to `TT-RNN`. The target function we are approximating is the state transition function $f() = f(\mathbf{s} \otimes \cdots \otimes \mathbf{s})$. We can express the function using FTT, followed by the polynomial expansion of the states concatenation $P_{TT}$. The approximation error of `TT-RNN`, viewed as one layer hidden

$$
\begin{aligned}
\|f - P_{TT}\| &\leq \|f - f_{TT}\| + \|f_{TT} - P_{TT}\| \\
&\leq \|f\| \sqrt{(d-1) \frac{(r+1)^{-(k-1)}}{(k-1)}} + C(k) p^{-k} |f_{TT}|_k \\
&\leq \|f - f_n\| \sqrt{(d-1) \frac{(r+1)^{-(k-1)}}{(k-1)}} + C(k) p^{-k} \sum_{i=k} \|D^{(i)}(f_{TT} - f_n)\| + o(\|f_n\|) \\
&\leq \frac{C_f^2}{\sqrt{n}} \left( \sqrt{(d-1) \frac{(r+1)^{-(k-1)}}{(k-1)}} + C(k) p^{-k} \sum_{i=k} \|D^{(i)} f_{TT}\| \right) + o(\|f_n\|)
\end{aligned}
$$

Where $p$ is the order of tensor and $r$ is the tensor-train rank. As the rank of the tensor-train and the polynomial order increase, the required size of the hidden units become smaller, up to a constant that depends on the regularity of the underlying dynamics $f$.

## 7.2 Training and Hyperparameter Search

We trained all models using the RMS-prop optimizer and employed a learning rate decay of $0.8$ schedule. We performed an exhaustive search over the hyper-parameters for validation. Table 2 reports the hyper-parameter search range used in this work.

| Hyper-parameter search range | | | |
|---|---|---|---|
| learning rate | $10^{-1} \ldots 10^{-5}$ | hidden state size | $8, 16, 32, 64, 128$ |
| tensor-train rank | $1 \ldots 16$ | number of lags | $1 \ldots 6$ |
| number of orders | $1 \ldots 3$ | number of layers | $1 \ldots 3$ |

Table 2: Hyper-parameter search range statistics for `TT-RNN` experiments.

For all datasets, we used a $80\% - 10\% - 10\%$ train-validation-test split and train for a maximum of $1e^4$ steps. We compute the moving average of the validation loss and use it as an early stopping criteria. We also did not employ scheduled sampling, as we found training became highly unstable under a range of annealing schedules.

## 7.3 DATASET DETAILS

**Genz** Genz functions are often used as basis for evaluating high-dimensional function approximation. In particular, they have been used to analyze tensor-train decompositions (Bigoni et al., 2016). There are in total 7 different Genz functions. (1) $g_1(x) = \cos(2\pi w + cx)$, (2) $g_2(x) = (c^{-2} + (x + w)^{-2})^{-1}$, (3) $g_3(x) = (1 + cx)^{-2}$, (4) $e^{-c^2\pi(x-w)^2}$ (5) $e^{-c^2\pi|x-w|}$ (6) $g_6(x) = \begin{cases} 0 & x > w \\ e^{cx} & else \end{cases}$ . For each function, we generated a dataset with $10,000$ samples using (2) with $w = 0.5$ and $c = 1.0$ and random initial points draw from a range of $[-0.1, 0.1]$.

**Traffic** We use the traffic data of Los Angeles County highway network collected from California department of transportation `http://pems.dot.ca.gov/`. The dataset consists of 4 month speed readings aggregated every 5 minutes . Due to large number of missing values ($\sim 30\%$) in the raw data, we impute the missing values using the average values of non-missing entries from other sensors at the same time. In total, after processing, the dataset covers $35\,136$, time-series. We treat each sequence as daily traffic of 288 time stamps. We up-sample the dataset every 20 minutes, which results in a dataset of $8\,784$ sequences of daily measurements. We select 15 sensors as a joint forecasting tasks.

**Climate** We use the daily maximum temperature data from the U.S. Historical Climatology Network (USHCN) daily (`http://cdiac.ornl.gov/ftp/ushcn_daily/`) contains daily measurements for 5 climate variables for approximately 124 years. The records were collected across more than $1\,200$ locations and span over $45\,384$ days. We analyze the area in California which contains 54 stations. We removed the first 10 years of day, most of which has no observations. We treat the temperature reading per year as one sequence and impute the missing observations using other non-missing entries from other stations across years. We augment the datasets by rotating the sequence every 7 days, which results in a data set of $5\,928$ sequences.

We also perform a DickeyFuller test in order to test the null hypothesis of whether a unit root is present in an autoregressive model. The test statistics of the traffic and climate data is shown in Table 3, which demonstrate the non-stationarity of the time series.

|  | **Traffic** | | **Climate** | |
|---|---|---|---|---|
| Test Statistic | 0.00003 | 0 | 3e-7 | 0 |
| p-value | 0.96 | 0.96 | 1.12 e-13 | 2.52 e-7 |
| Number Lags Used | 2 | 7 | 0 | 1 |
| Critical Value (1%) | -3.49 | -3.51 | -3.63 | 2.7 |
| Critical Value (5%) | -2.89 | -2.90 | -2.91 | -3.70 |
| Critical Value (10%) | -2.58 | -2.59 | -2.60 | -2.63 |

Table 3: Dickey-Fuller test statistics for traffic and climate data used in the experiments.

## 7.4 PREDICTION VISUALIZATIONS

Genz functions are basis functions for multi-dimensional Figure 10 visualizes different Genz functions, realizations of dynamics and predictions from TLSTM and baselines. We can see for "oscillatory", "product peak" and "Gaussian ", TLSTM can better capture the complex dynamics, leading to more accurate predictions.

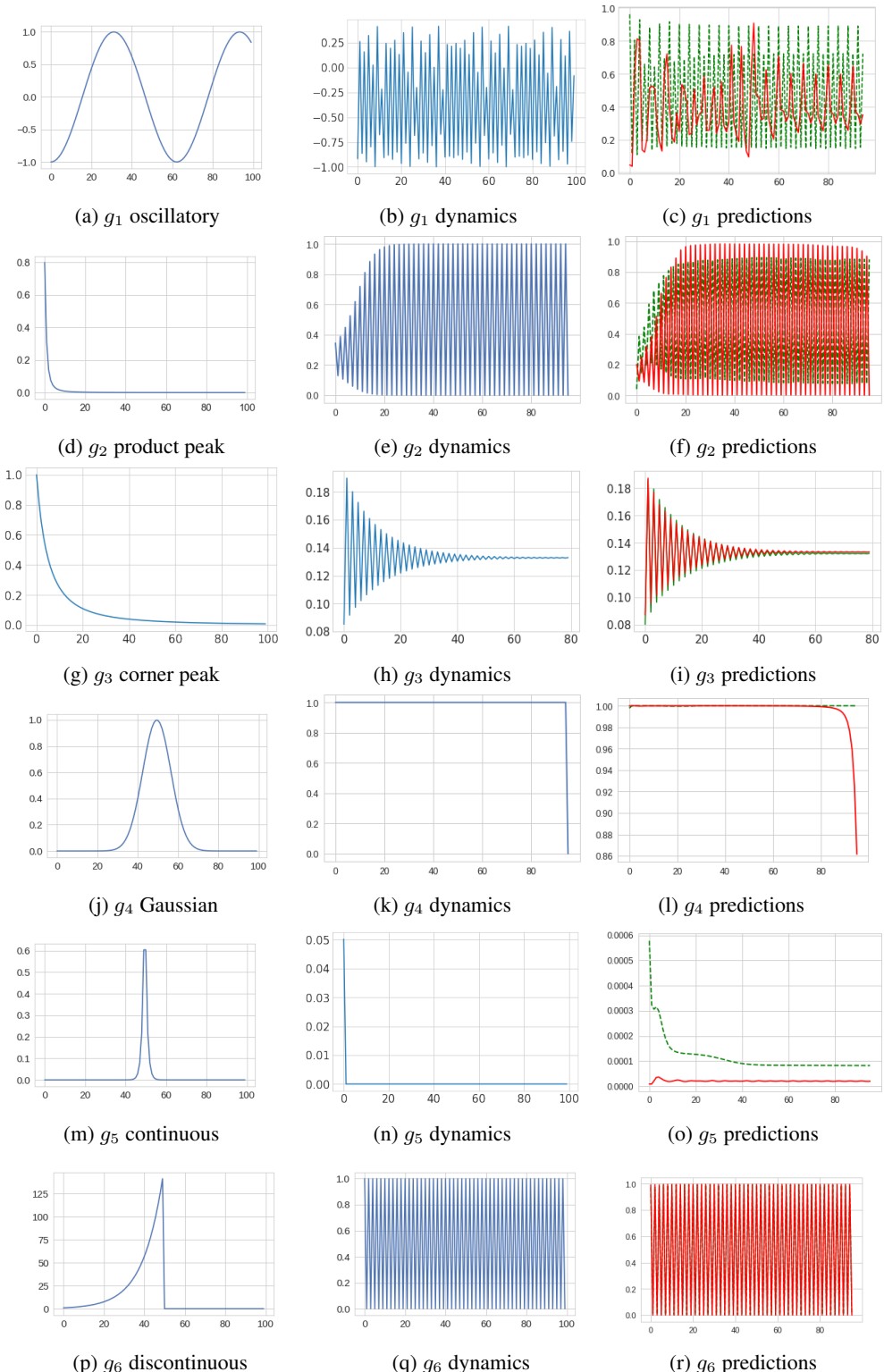

Figure 10: Visualizations of Genz functions, dynamics and predictions from TLSTM and baselines. Left column: transition functions, middle: realization of the dynamics and right: model predictions for LSTM (green) and TLSTM (red).

### 7.5 MORE CHAOTIC DYNAMICS RESULTS

Chaotic dynamics such as Lorenz attractor is notoriously different to lean in non-linear dynamics. In such systems, the dynamics are highly sensitive to perturbations in the input state: two close points can move exponentially far apart under the dynamics. We also evaluated tensor-train neural networks on long-term forecasting for Lorenz attractor and report the results as follows:

**Lorenz**   The Lorenz attractor system describes a two-dimensional flow of fluids (see Figure 9):

$$\frac{\mathrm{d}x}{\mathrm{d}t} = \sigma(y-x), \quad \frac{\mathrm{d}y}{\mathrm{d}t} = x(\rho-z) - y, \quad \frac{\mathrm{d}z}{\mathrm{d}t} = xy - \beta z, \quad \sigma = 10, \rho = 28, \beta = 2.667.$$

This system has chaotic solutions (for certain parameter values) that revolve around the so-called Lorenz attractor. We simulated $10\,000$ trajectories with the discretized time interval length $0.01$. We sample from each trajectory every $10$ units in Euclidean distance. The dynamics is generated using $\sigma = 10$ $\rho = 28$, $\beta = 2.667$. The initial condition of each trajectory is sampled uniformly random from the interval of $[-0.1, 0.1]$.

Figure 11 shows 45 steps ahead predictions for all models. HORNN is the full tensor `TT-RNN` using vanilla RNN unit without the tensor-train decomposition. We can see all the tensor models perform better than vanilla RNN or MRNN. `TT-RNN` shows slight improvement at the beginning state.



|     (a) RNN     |     (b) MRNN     |     (c) HORNN     |     (d) TT-RNN     |     (e) TLSTM     |

Figure 11: Long-term (right 2) predictions for different models (red) versus the ground truth (blue). `TT-RNN` shows more consistent, but imperfect, predictions, whereas the baselines are highly unstable and gives noisy predictions.

