# OpenReview forum: "Long-term Forecasting using Tensor-Train RNNs"
_ICLR.cc/2018/Conference — Reject_

### Official Review · AnonReviewer3 · 2017-11-24
**Needs work on presentation and more thorough comparisons with other methods.**

**Rating:** 4
**Confidence:** 4

**Review:**

The paper proposes Tensor-Train RNN and Tensor-Train LSTM (TT-RNN/TLSTM), a RNN/LSTM architecture whose hidden unit at time t h_t is computed from the tensor-vector product between a tensor of weights and a concatenation of hidden units from the previous L time steps. The motivation is to incorporate previous hidden states and high-order correlations among them to better predict long-term temporal dependencies for seq2seq problems. To address the issue of the number of parameters growing exponentially in the rank of the tensor, the model uses a low rank decomposition called the ‘tensor-train decomposition’ to make the number of parameters linear in the rank. Some theoretical analysis on the number of hidden units required for a given estimation error, and experimental results have been provided for synthetic and real sequential data.

First of all, the presentation of the method in section 2.1 is confusing and there seem to be various ambiguities in the notation that harms understanding of the method. The tensor-vector product in equation (6) appears problematic. The notation that I think is standard is as follows: given a tensor W \in R^{n_1 \times … \times n_P} and vectors v_p \in R^{n_p}, the tensor-vector product W \times_{p=1}^P v_p = vec(W) \otimes_{p=1}^P v_p = \sum{i_1,...,i_P} \prod_{p=1}^P v_{p,i_p}. So I’m guessing you want to get rid of the \otimes signs (the kronecker products) in (6) or you want to remove the summation and write W \times_{p=1}^P s_{t-1}. Also \alpha that appears in (6) is never defined. Is it another index? This is confusing because you say W is P-dimensional but have P+1 indices for it including alpha (W_{\alpha i_1 … i_p}). Moreover the dimensionality of W^{hx} x_t in (6) is R^H judging from the notation in page 2, but isn’t the tensor-vector product a scalar? Also am I correct in thinking that s_{t-1} should be [1, h_{t-1}^T, …, h_{t-L}^T], i.e. a vector of length LH+1 rather than a matrix? The notation from page 2 implies that you are using column vectors, so the definition of s_{t-1} makes it appear as an (L+1) by H matrix, which could make the reader interpret s_{t-1;i_1} in (6) as vectors instead of scalars (this is reinforced by the kronecker product between these s_{t-1;i_p}). I had to work this out from the number of parameters (HL+1)^P in section 2.2. The diagram of s_{t-1} in Figure 3 is also confusing, because it isn’t obvious that the unlabelled grey bars are copies of s_{t-1}. Also I notice that the name ‘Tensor Train RNN/LSTM’ has been used in Yang et al, 2017. You probably want to avoid using the same name since the models are different. It would be nice if you could explain in a bit more detail about how they are different in the related work section.

Assuming I have understood the method correctly, the idea of using tensor products to incorporate higher order interactions between the hidden states at different times appears sensible. From the theoretical analysis, you claim that 1) smoother f is easier to approximate, and 2) polynomial interactions are more efficient than linear ones. The first point seems fairly self-explanatory and doesn’t seem to require a proof. The second point isn’t so convincing because you have two additive terms on the right hand side of the inequality in Theorem 3.1 (btw I’m guessing you want the inequality to be the other way round): the first term is independent of p, and the second decreases exponentially with p. Your second point would only hold if this first term is reasonably small, but this doesn’t seem obvious to me.

Regarding the experiments, I’m sceptical as to whether a grid search over hyperparameters for TLSTM vs grid search over the same hyperparameters for (M)LSTM provides a fair comparison. You probably want to compare the models given the same number of parameters, since given the same state size, TLSTM will have many more parameters than (M)LSTM. A plot of x-axis: # parameters, y-axis: average RMSE at convergence would be informative. Moreover for figure 8, you probably want to control the time taken for training instead of just comparing validation loss at the same number of steps. I imagine the best performing TLSTM model will have many more parameters and hence take much longer to train than the best performing LSTM model.
Moreover, it seems as though the increased prediction accuracy from LSTM is marginal considering you have 3 more hyperparameters to tune (L,S,P - what was the value of P used for the experiments?) and that tuning them is important to prevent overfitting.

I’m also curious as to how TLSTM compares to hierarchical RNN approaches for modelling long-term dependencies. It will be interesting to compare against models like Stacked LSTM (Graves, 2013), Grid LSTM (Kalchbrenner, 2015) and HM LSTM (Chung, 2017). These models have mostly been evaluated on text, but I don’t see any reason they can’t be extended to sequential forecasting on time series data. Also regularisation techniques such as batch-norm for LSTMs (Cooijmans et al, 2016) and layer-norm (Ba et al, 2016) seem to help a lot for increasing prediction accuracy. Did you investigate these techniques to control overfitting?

Other minor comments on presentation:
For figure 6, the legends are inconsistent with the caption. Also you might want to overlay predictions on top of the ground truth for better comparison and also to save space.

Overall, I think there are vast scopes for improvement in presentation and comparisons with other methods, and hence find the paper not yet ready for publication.

---

### Official Review · AnonReviewer2 · 2017-11-27
**This paper propose an RNN architecture that takes the last L-step history to parameterize the latent state. The parameterization takes the form of P-degree polynomial interaction between last L hidden states. Factorization method is used to tackle the dimensionality issue. Experimental comparisons over synthetic and real-world datasets show advantages over traditional LSTM.**

**Rating:** 5
**Confidence:** 3

**Review:**

For method:
though it is known that RNN lacks the ability to capture long term dependency, it is designed to take infinite order of history (e.g., dimension of h is large enough, or f(x, h_t-1) is flexible enough). So the claim that RNN only learns a Markov Model is improper. For example, in “Recurrent Marked Temporal Point Processes: Embedding Event History to Vector”, it shows that RNN has the ability to fit the intensity function of Hawkes process (which has infinite order dependency).

Decomposing tensor operator as a layer in neural network is not new. For example, “Tensor Contraction Layers for Parsimonious Deep Nets”. The technique used in this paper is tensor-trains, which is also proposed previously.

Also the author only talked about # parameters. A more important issue is the time cost. The author should also explicitly analyze the computation cost.

For writing:

Some sections needs more explanations. In Section 2.1, it seems S_{t-1} is an (1 + L x H)-dimensional vector, according to your definition. Then how is S_{t-1; i_1} defined? Figure 2 has little information about proposed architecture. While Figure 3 is also very vague.

Notations are not quite consistent. In Figure 3, the S_{t-1} contains K history vectors. What is K here? I suppose it is the same as L. In Figure 6, the legend says red curve is LSTM, but the caption says the green one is LSTM.

For experiment:

The datasets used are small, with a few number of not quite long sequences. But for demonstration purpose this might be ok. The doubt is whether this method is scalable to large datasets? Analysis like time cost, memory consumption needs to be included, in order for people to get an idea of its scalability.

Figure 8 shows the convergence. I would say the difference is not significant. Consider its computation cost, I would doubt the ‘much faster’ claim in Page 7.

Also it seems the proposed method has more parameters than traditional RNN. To get a fair comparison, higher dimensionality of latent state should be used in LSTM.

Overall the paper tries to tackle an important problem, which is good. However, both methods and experiments need improvement.

---

### Official Review · AnonReviewer1 · 2017-11-27
**Proposes Tensor-Train RNN and provides some (flawed) experiments showing it’s better at long-term forecasting than RNN and LSTM.**

**Rating:** 6
**Confidence:** 4

**Review:**

This work addresses an important and outstanding problem: accurate long-term forecasting using deep recurrent networks.  The technical approach seems well motivated, plausible, and potentially a good contribution, but the experimental work has numerous weaknesses which limit the significance of the work in current form.

For one, the 3 datasets tested are not established as among the most suitable, well-recognized benchmarks for evaluating long-term forecasting.  It would be far more convincing if the author’s used well-established benchmark data, for which existing best methods have already been well-tuned to get their best results.  Otherwise, the reader is left with concerns that the author’s may not have used the best settings for the baseline method results reported, which indeed is a concern here (see below).

One weakness with the experiments is that it is not clear that they were fair to RNN or LSTM, for example, in terms of giving them the same computation as the TT-RNNs. Section Hyper-parameter Analysis” on page 7 explains that they determined best TT rank and lags via grid search.  But presumably larger values for rank and lag require more computation, so to be fair to RNN and LSTM they should be given more computation as well, for example allowing them more hidden units than TT-RNNs get, so that overall computation cost is the same for all 3 methods.  As far as this reviewer can tell, the authors offer no experiments to show that a larger number of units for RNN or LSTM would not have helped them in improving long-term forecasting accuracies, so this seems like a very serious and plausible concern.

Also, on page 6 the authors say that they tried ARMA but that it performed about 5% worse than LSTM, and thus dismissing direct comparisons of ARMA against TT-RNN.  But they are unclear whether they gave ARMA as much hyper-parameter tuning (e.g. for number of lags) via grid search as their proposed TT-RNN benefited from.  Again, the concern here is that the experiments are plausibly not being fair to all methods equally.

So, due to the weaknesses in the experimental work, this work seems a bit premature.  It should more clearly establish that their proposed TT-RNN are indeed performing well compared to existing SOTA.

---

### Decision · Program_Chairs · 2018-01-29
**ICLR 2018 Conference Acceptance Decision**

**Decision:**

Reject

**Comment:**

This paper address the increasingly studied problem of predictions over long-term horizons. Despite this, and the important updates from the authors, the paper is not yeat ready and improvements identified include more control over the fair comparisons, improved clarity in exposition.